# Building an Ethical and Trustworthy Biomedical AI Ecosystem for the Translational and Clinical Integration of Foundation Models

**DOI:** 10.3390/bioengineering11100984

**Published:** 2024-09-29

**Authors:** Baradwaj Simha Sankar, Destiny Gilliland, Jack Rincon, Henning Hermjakob, Yu Yan, Irsyad Adam, Gwyneth Lemaster, Dean Wang, Karol Watson, Alex Bui, Wei Wang, Peipei Ping

**Affiliations:** 1Department of Physiology, University of California, Los Angeles, CA 90095, USA; sankarb@g.ucla.edu (B.S.S.); dcg5438@psu.edu (D.G.); jmrincon@g.ucla.edu (J.R.); yuyan666@g.ucla.edu (Y.Y.); irsyadadam@g.ucla.edu (I.A.); gwynethsage@gmail.com (G.L.); dingwang@g.ucla.edu (D.W.); 2NIH CFDE ICC-SC, NIH BRIDGE2AI Center & NHLBI Integrated Cardiovascular Data Science Training Program, UCLA, Los Angeles, CA 90095, USA; 3European Molecular Biology Laboratory, European Bioinformatics Institute (EMBL-EBI), Cambridge CB10 1SD, UK; hhe@ebi.ac.uk; 4Bioinformatics IDP, University of California, Los Angeles, CA 90005, USA; 5Department of Medicine, Cardiology Division, University of California, Los Angeles, CA 90095, USA; kwatson@mednet.ucla.edu; 6Medical Informatics Home Area, University of California, Los Angeles, CA 90095, USA; buia@mii.ucla.edu; 7Department of Computer Science, University of California, Los Angeles, CA 90095, USA

**Keywords:** biomedical AI, Foundation Models, AI ecosystem, AI lifecyle, clinical integration, ethical AI, trustworthy AI, AI governance and regulation, stakeholder engagement

## Abstract

Foundation Models (FMs) are gaining increasing attention in the biomedical artificial intelligence (AI) ecosystem due to their ability to represent and contextualize multimodal biomedical data. These capabilities make FMs a valuable tool for a variety of tasks, including biomedical reasoning, hypothesis generation, and interpreting complex imaging data. In this review paper, we address the unique challenges associated with establishing an ethical and trustworthy biomedical AI ecosystem, with a particular focus on the development of FMs and their downstream applications. We explore strategies that can be implemented throughout the biomedical AI pipeline to effectively tackle these challenges, ensuring that these FMs are translated responsibly into clinical and translational settings. Additionally, we emphasize the importance of key stewardship and co-design principles that not only ensure robust regulation but also guarantee that the interests of all stakeholders—especially those involved in or affected by these clinical and translational applications—are adequately represented. We aim to empower the biomedical AI community to harness these models responsibly and effectively. As we navigate this exciting frontier, our collective commitment to ethical stewardship, co-design, and responsible translation will be instrumental in ensuring that the evolution of FMs truly enhances patient care and medical decision-making, ultimately leading to a more equitable and trustworthy biomedical AI ecosystem.

## 1. Introduction

A corollary to the rise of “Big Data” is the development of large-scale machine learning models that have the capacity to learn from large datasets [1]. FMs are large-scale models that can be trained on large-scale datasets and serve as the “foundation” for downstream tasks related to the original model. They are increasingly recognized as a component in the workflow for large-scale AI development, leveraging millions to billions of parameters through self-supervised, unsupervised, or semi-supervised learning techniques [2]. The initial training of FMs on large datasets enables them to learn patterns, structures, and context within the data without the need for labor-intensive manual annotation. This initial pre-training provides a foundation for further adaptation and fine-tuning across diverse tasks, spanning from predictive analytics to generative applications [3].

Biomedical AI technologies have shown promising capabilities to diagnose, predict, and recommend treatments across a variety of medical modalities and data types, such as electronic health records (EHRs), chest X-rays, and electrocardiograms [4]. With FMs achieving state-of-the-art performance in natural language processing (NLP) and computer vision (CV), there is growing interest in exploring their utility in biomedical applications. Notably, pre-trained FM architectures can be fine-tuned for various downstream biomedical tasks [4]. For example, BioLinkBERT, pre-trained using the Bidirectional Encoder Representations from Transformers (BERT) architecture on citation-linked biomedical corpora from PubMed, demonstrated utility for downstream biomedical natural language processing (BioNLP) tasks such as named entity recognition, document classification, and question answering [5]. Another example is HeartBEiT, an FM pre-trained on 12-lead ECG image data and fine-tuned for the classification of patients with reduced left ventricular ejection fraction and the classification of patients with hypertrophic cardiomyopathy [6]. Lastly, scFoundation, a model pre-trained on over 50 million scRNA-seq data using an encoder–decoder transformer, has shown its utility in a diverse array of single-cell analysis tasks such as gene expression enhancement, tissue drug response prediction, and single-cell drug response classification [7].

Looking forward, FMs pre-trained on multimodal biomedical datasets hold promise for generalizing and integrating knowledge across various data types, learning new tasks dynamically, and addressing a wide array of medical challenges [8]. However, this promising concept for biomedical AI also presents unique ethical challenges, necessitating heightened vigilance and responsible development. Adopting an ethically governed, co-designed approach to FM development and clinical integration, grounded in evidence-based principles and prioritizing the needs of impacted individuals and communities, is critical. This AI pipeline should allow for continuous refinement of AI technologies based on AI stewardship, which is the feedback from stakeholders and regulatory entities. Such an approach is essential to ensure that AI innovations positively contribute to healthcare and public health, reinforcing necessary safeguards and ethical standards.

The main contributions of this paper are as follows (Figure 1):We outline the essential components of an AI Ecosystem for integrating biomedical AI technologies into clinical and/or translational settings;We examine the current landscape of ethical considerations and ethical practices broadly applicable to biomedical AI and with applications to FM development and deployment, highlighting critical challenges and existing mitigation strategies across three key areas:
Mitigating Bias and Enhancing FairnessEnsuring Trustworthiness and ReproducibilitySafeguarding Patient Privacy and Security;We examine pivotal government and scholarly publications that chart the present guidelines and future directions for AI stewardship, emphasizing two main components:
AI Governance and RegulationStakeholder Engagement; Finally, we discuss a unified perspective of how the principles of ethical and trustworthy AI and stewardship in AI integrate into the ecosystem.

## 2. AI Ecosystem in Biomedicine

The AI ecosystem is a concept that defines the complex interdependent patterns that connect developers, users, and the upstream and downstream resources necessary for AI development and deployment. It provides a structure to develop an ethical and regulatory framework that promotes fairness, transparency, and accountability in the development and use of AI/ML systems [9]. Within the AI biomedical ecosystem, a well-defined AI pipeline guided by AI stewardship drives the direction of model development toward clinical and translational integration while ensuring its responsible and ethical use. The pipeline begins with the management of large biomedical datasets and culminates in a thoroughly validated model. However, the pipeline workflow is amenable to moving forward or backward, as dictated by AI stewardship. To clarify, the AI ecosystem can best be defined via nine key components presented as a pipeline in Figure 2 [10,11,12]:-Data Lifecycle Management: The collection, dissemination, and curation of vast amounts of diverse biomedical data.-Data Repositories: Centralized systems that store, validate, and distribute data, promoting transparent and reproducible AI technologies.-Data Processing: Cleaning, annotating, and structuring data to make them AI-ready. Model Development: The development and training of FMs that can then be utilized for various downstream tasks such as hypothesis generation, explanation, causal reasoning, and clinical decision support.-Model Repositories: Centralized storage for managing and sharing AI models to promote accessibility and collaboration amongst stakeholders. Centralized storage for managing and sharing AI models to promote accessibility and collaboration amongst stakeholders.-Model Evaluation: The assessment of model performance and reliability prior to deployment in biomedical settings.-Clinical Translation: Operationalizing FMs in a clinical setting to enhance patient care.-AI Governance and Regulation: Established legal and ethical standards that enforce compliance through both government processes and committee/board regulation.-Stakeholder Engagement: Diverse communities and individuals contributing to and affected by AI in biomedicine. Engagement refers to the active participation of these groups in the entire pipeline of AI from bench to bedside in both an industrial and healthcare setting.

## 3. Ethical Considerations in the AI Pipeline for Foundation Models

### 3.1. Mitigating Bias and Enhancing Fairness

Bias in AI can refer to two distinct concepts: technical bias and social bias. Technical bias refers to a statistical concept related to model assumptions that are made for ease of learning and generalization but that introduce error. This technical bias can lead to underfitting, where the model, due to its oversimplified assumptions, fails to capture the complexity of the data. Social bias in AI refers to the prejudices reflected in the outputs of AI systems, often due to biases present in the training data. These biases often mirror social biases, including historical and current social inequalities. Social biases in biomedical AI can cause direct social harm when they perpetuate outdated claims, lead to inaccurate insights, compromise the quality of care for marginalized groups, and/or exacerbate disparities in the quality of care. AI Fairness is the practice of seeking to understand and mitigate these social biases. As summarized in Table 1, we identify key social biases present in biomedical data and review evidence-based techniques to mitigate them. In Section 3.1.1 and Section 3.1.2, we investigate the social biases that profoundly influence the ethics and trustworthiness of biomedical FM applications. We specifically discuss biases arising from the underrepresentation of certain demographic groups in biomedical datasets and the stereotypical biases inherent in natural language data used to train certain biomedical FMs, notably Language Model Large (LLMs) and Vision Language Models (VLMs). For each type of social bias, we explore mitigation strategies aimed at neutralizing their negative impacts.

Table 1. We explored social biases present in biomedical data that affect the fairness of biomedical AI. Subsequently, we identified evidence-based techniques to mitigate these common challenges. A more detailed explanation of these concepts and relationships is described in Section 3.1.1 and Section 3.1.2. (Abbreviations: TP—True Positive, FP—False Positive).

#### 3.1.1. Social AI Bias: Underrepresentation Bias

Social bias in AI can arise when the training data do not accurately represent the real-world distribution of labels and/or features [40]. In biomedicine, this social bias manifests as an underrepresentation of demographic groups in clinical trials and biomedical data registries when compared with the distribution of the general population [41,42,43]. If biomedical datasets lack sufficient representation from all demographic subgroups, AI models trained on such data may not effectively capture each group’s specific feature distribution. Consequently, these models may exhibit poor generalization performance when making decisions for individuals from these underrepresented groups. Therefore, in the context of healthcare delivery, downstream prediction tasks in FM development may exhibit disparities in prediction metrics across different subgroups. When these AI systems are deployed, they can lead to unfair outcomes for certain subgroups as well as perpetuate or exacerbate health inequities [9,44]. For example, large biorepositories supporting omics datasets predominantly consist of data from individuals of European descent, leading to a notable underrepresentation of racial and ethnic minorities that impacts equitable healthcare delivery in precision medicine applications [42,43].

First, we examine methods for mitigating underrepresentation bias at the data level. It is crucial that we work towards the inclusive collection of biomedical datasets encompassing a diverse range of demographic groups [44]. This includes a proportional representation of underrepresented groups based on race/ethnicity, gender, socioeconomic status, age, disability status, geography, and other characteristics. However, inclusivity in data collection is a long-term goal that requires tremendous mobilization of resources and amending of historical mistrust among certain demographic populations in biomedical research participation. In the short term, foundation AI model providers can alleviate the issue of imbalanced representation by publishing data sheets or model cards that specify data sources and demographic breakdowns used during model development. This transparency allows users to fine-tune pre-trained models with their own datasets, helping to better balance representation across demographic groups in real-world applications. In conjunction with these efforts, synthetic datasets present a potential strategy to augment underrepresented groups, improving representation during pre-training and fine-tuning [14]. Synthetic data, artificially generated through computer simulations or algorithms, attempt to closely mirror the statistical properties of real-world data. It is a broad concept that encompasses a variety of processes and techniques, from techniques that transform the data to advanced deep learning techniques that generate data by learning from real-world data. In biomedicine, synthetic datasets can be engineered to reduce bias by closely emulating the phenotypes of underrepresented individuals, allowing the training of biomedical AI models that can generalize across demographic groups [14]. Nonetheless, the use of synthetic data introduces challenges such as potential inaccuracies, noise, over-smoothing, and inconsistencies when compared to real-world data [15]. Therefore, while synthetic datasets provide a potential solution to address underrepresentation, careful consideration must be given to their generation and use to ensure they contribute positively to the development of fair and unbiased AI systems.

Second, we examine mitigation strategies that can be applied during model training to address the unfairness in downstream FM prediction tasks caused by data under-representation bias. These strategies can be applied to various data types but necessitate datasets to have associated demographic metadata. One approach includes importance weighting, a technique where samples from underrepresented groups in the dataset are shown more frequently to the model, giving them higher importance in model decision-making [16,17]. Another approach involves the use of adversarial strategies to address bias in machine learning models. In machine learning, an adversary refers to an agent that intentionally seeks to deceive, manipulate, or exploit a model to achieve a specific goal; it has been explored as a potential strategy to both mitigate and detect bias [18,45]. For instance, Yang et al. introduce a novel adversarial debiasing framework to determine COVID-19 status while mitigating biases [19]. By enhancing the loss function for better model convergence, they demonstrate its effectiveness in debiasing patient ethnicity and hospital location, addressing multiclass sensitive features—a significant advancement over prior binary-focused methods. An additional strategy is regularization, which is commonly applied in prediction tasks to reduce overfitting but can also be adapted to reduce disparities across different demographic groups in the model’s predictions. This is achieved by introducing a regularization hyperparameter in the model’s loss function that penalizes model weights resulting in high values of a chosen fairness metric [20,21]. Lastly, dropout regularization, which applies a binary mask to neurons so that each neuron has a probability (p) of being dropped out, can also reduce unfair predictions by decreasing the model’s reliance on specific sensitive demographic features [22]. It is important to note that each method to improve model fairness may come with potential trade-offs in performance and training time.

For FM downstream tasks, it is imperative to assess the model’s fairness, particularly when underrepresentation bias is present. This evaluation should consider the uncertainties of bias mitigation at both the dataset and training levels to ensure the model performs equitably in real-world applications. One common approach to assessing fairness in classification tasks is by evaluating the impact of sensitive demographic features on model predictions using fairness metrics. For instance, equalized odds measure the classifier’s accuracy with regard to the true positive rates and false positive rates between different demographics [23,24]. Similarly, the equal opportunity metric requires that the true positive rates are equal for all demographic groups, ensuring that all groups equally benefit from the model [24,25]. On the other hand, predictive parity ensures that precision, or the likelihood of a true positive, is equal across different demographic groups, meaning individuals receiving the same decision should have equal outcomes [26]. While these metrics can provide insight into how the model’s decisions may be disparate or equal among demographic groups, relying on a single fairness metric can be limiting; it may provide an incomplete or misleading picture of the model’s social bias since each metric captures only a specific aspect of fairness. In addition, these metrics are limited to classification tasks. Therefore, evaluating a model’s dependence on sensitive demographic features or its disregard for socio-demographic variables can be achieved using explainability techniques (see Section 3.2.2). These techniques help elucidate the relationship between input data attributions and model outputs, ensuring that AI in biomedicine is fair and equitable [27]. However, applying explainability techniques comes with challenges. These methods can be resource-intensive, and their effectiveness often depends on human interpretation. Moreover, explainability techniques can offer either a global understanding of how features influence the overall decision-making process or a local understanding of how features affect individual decisions [46]. While both perspectives are valuable, each provides only a partial view of the model’s behavior.

#### 3.1.2. Social AI Bias: Stereotypical Biases

Stereotypical biases in real-world datasets are a significant concern in the field of AI. These biases, often reflections of societal stereotypes, can be unintentionally incorporated into AI models during the training process (Figure 3). Historical and persistent stereotypical biases are particularly prevalent in natural language data, making pre-trained LLMs—a type of FM trained on extensive text corpora for a variety of downstream NLP tasks—prone to learning and perpetuating these biases [47]. The presence of stereotypical biases can lead to unfair outcomes when LLMs are used in applications such as text classification, sentiment analysis, or recommendation systems. Given the potential for LLMs in clinical settings for tasks like compiling patient notes and aiding in clinical decision-making, the development and deployment of biomedical LLMs is tempered by ethical concerns [48,49]. Biomedical LLMs can learn and even amplify stereotypical biases present in the biomedical text corpora. When examining four major commercial LLMs (Bard, ChatGPT-3.5, Claude, GPT-4) on medical queries related to race-based practices, these models often perpetuated outdated, debunked stereotypes from their biased training data [49]. Additionally, a UNESCO study on AI gender biases provided clear evidence of gender stereotyping across various LLMs, emphasizing the need for systematic changes to ensure fairness in AI-generated content [50]. This highlights the presence of stereotype biases in LLMs used in biomedical AI systems and underscores the need to address these biases across all demographic groups, including by race and gender.

Recognizing the critical need to address stereotypical biases inherent in natural language data, a variety of specific strategies have been investigated to mitigate these biases in LLMs. One such strategy is counterfactual data augmentation (CDA), where demographic identifiers like ‘he’ and ‘she’ are swapped to balance the dataset [29,30]. This method helps to reduce the reinforcement of existing stereotypes by providing a diverse set of examples, ultimately leading to fairer model outputs. Counterfactual data augmentation, while useful, may not adequately address more complex biases that are not directly associated with demographic identifiers. In their work, ‘Queens are Powerful too: Mitigating Gender Bias in Dialogue Generation,’ the authors propose bias control training to mitigate gender bias in generative dialogue models [31]. By associating control tokens with gendered properties, they achieve granular control over the “genderedness” of model outputs while maintaining dialogue quality and safety [31]. However, this process of bias control training is highly dependent on the categorization of responses regarding their stereotype associations. Complimenting bias control training includes minimizing bias in word embeddings, which are vector representations that capture the semantic and syntactic properties of words. As the foundation of LLMs, word embeddings enable these models to understand and generate human-like text. Word embedding biases refer to the phenomenon where these embeddings reflect and perpetuate societal biases present in the training data [51]. This means that words associated with a particular protected group (e.g., gender) might be closer in the embedding space to stereotypically associated words. To reveal biases related to protected groups, one approach involves identifying pairs of data points that differ in a specific attribute, creating a “seed direction” that represents this difference [32]. Analogous pairs are then generated and scored based on their alignment with the seed direction, systematically uncovering biased relationships in the embeddings. Once identified, a method to debias these words involves generating augmented sentences through demographic identifier word swapping, encoding both the original and augmented sentences and maximizing their mutual information [33]. However, the process of debiasing word embeddings can encounter difficulties in maintaining semantic consistency. This presents a decision-making challenge where one must balance performance trade-offs when implementing debiasing techniques. Ma et al. explored how attention heads can encode bias and found that a small subset of attention heads within pre-trained language models are primarily responsible for encoding stereotypes toward specific minority groups and could be identified using attention maps [32]. The authors used Shapley values [52] to estimate the contribution of each attention head to stereotype detection and performed ablation experiments to assess the impact of pruning the most and least contributive heads [34]. Attention head pruning, while effective in model compression and mitigating encoded stereotypes, is a technique specific to architectures that utilize attention heads. It may also risk the loss of semantic meaning that can impact the model’s performance.

Stereotypical biases inherent in natural language data can also influence multimodal models that incorporate natural language, such as VLMs [53]. These biases, if unaddressed, can inadvertently affect the performance and fairness of these models. For example, VLMs like DALL-E and Midjourney have been shown to exhibit racial and stereotypical biases in their outputs [9]. For instance, when prompted to generate images of CEOs, these models predominantly produced images of men, reflecting gender bias acquired during training. Saravanan et al. also explored social bias in text-to-image foundation models performing image editing tasks [54]. Their findings revealed significant unintended gender alterations, with images of women altered to depict high-paid roles at a much higher rate (78%) than men (6%). Additionally, there was a notable trend of skin lightening in images of Black individuals edited into high-paid roles [54]. However, VLMs in biomedicine can analyze visual and textual medical data for tasks such as medical report generation and visual question answering [55]. Biomedical VLMs can learn stereotypical associations between words and images, perpetuating these in their inference and exacerbating health inequities. Therefore, it is essential to explore methods that can disentangle the skewed similarities in the representation of certain images and their associated demographic annotations. Seth et al. introduce a debiasing framework for pre-trained vision-language models (VLMs) using an Additive Residual Learner (ARL) to disentangle protected attributes from image representations [35]. By training the ARL to modify image representations and reduce certainty in classifying attributes like race, gender, and age, the framework improves fairness without sacrificing zero-shot predictive performance [35]. However, the ARL may overcompensate during the debiasing process, potentially flipping the similarity skew in the opposite direction for different demographic annotations. This can result in unintended biases being introduced, highlighting the need for careful calibration and evaluation of the model’s performance across various demographic groups.

To complement the discussed debiasing strategies, model alignment techniques offer a versatile set of strategies for mitigating stereotypical biases in AI systems. These techniques aim to align models more closely with human values and preferences, enhancing their safety, fairness, and contextual appropriateness [36,56]. In the biomedical field, aligning LLMs and VLMs to avoid social stereotypes in decision-making is essential for creating more equitable and fairer AI. These alignment techniques are applicable to both LLMs and VLMs, enhancing their performance and adaptability. However, the specific methodologies and considerations may vary depending on the model architecture and the task at hand. Two foundational techniques in model alignment are Instruction Tuning (IT) and Supervised Fine-Tuning (SFT), each with distinct objectives. SFT involves fine-tuning model outputs against preferred outcomes [57] using a curated dataset of high-quality outputs. IT, on the other hand, fine-tunes models using a labeled dataset of instructional prompts and corresponding outputs [58], with the goal of improving performance on specific tasks and general instruction-following. These techniques should be used judiciously, keeping in mind the potential risks and challenges associated with fine-tuning, such as overfitting and distributional shifts.

Reinforcement Learning from Human Feedback (RLHF) offers another layer of refinement for LLMs by integrating human feedback into the learning process [37]. In reinforcement learning, an agent learns which actions to take by interacting with an environment, receiving rewards for correct actions and penalties for incorrect ones. RLHF enhances this process by incorporating human feedback to guide and accelerate learning, enabling the model to make more informed decisions. The primary goal of RLHF is to optimize a model’s responses based on a reward system that aligns with human preferences. For LLMs, this involves defining a policy that dictates responses to prompts and resulting completions. The process includes pre-training a base LLM, generating output pairs, and using a reward model (RM) that is trained on human feedback to mimic human ratings. The LLM is then trained to achieve high feedback scores from the RM. The challenge lies in optimizing these rewards without overly relying on them, as human preferences can be complex and nuanced. Reinforcement Learning with AI Feedback (RLAIF) [38] is a related approach that uses AI systems to evaluate actions and guide learning. In RLAIF, an AI system, such as an LLM, provides feedback instead of human evaluators. This feedback is used to train a reward model similar to RLHF but with AI-generated evaluations. The key difference between RLHF and RLAIF is in the source of feedback—human evaluators in RLHF versus AI systems in RLAIF. Another method, Direct Preference Optimization (DPO) [39], fine-tunes LLMs with human feedback without using reinforcement learning. Like RLHF, DPO involves generating output pairs and receiving human feedback, but the LLM is trained to assign high probabilities to positive examples and low probabilities to negative ones, effectively bypassing the need for reinforcement learning. However, DPO requires labeled positive and negative pairs for training, whereas RLHF, once the RM is trained, can annotate as much data as needed for fine-tuning.

Moreover, red teaming represents a way to assess the stereotypic outputs of LLMs and VLMs. It involves intentional adversarial attacks wherein an input is modified in a way that bypasses the model’s alignment to reveal inherent vulnerabilities, including biased output. This process often involves a human-in-the-loop, or another model, to assess and provoke the target model into producing harmful outputs. Red-teaming in biomedicine should engage multidisciplinary teams to evaluate AI systems and prevent biased medical information. For example, Chang et al. conducted a study using multidisciplinary red-teaming to test medical scenarios with adversarial commands, such as “you are a racist doctor” [59]. They exposed vulnerabilities in GPT-3.5 and 4.0 that allowed the propagation of identity-based discrimination and false stereotypes, influencing treatment recommendations and perpetuating discriminatory behaviors based on race or gender, such as biased renal function assessments. However, it is important to note that the red-teaming process may fail to expose certain biases if the adversarial inputs are not sufficiently diverse or comprehensive. Therefore, while valuable, red-teaming should be used alongside other strategies, such as continuous monitoring and training reward models on human interpretable objectives [60]. For example, Constitutional AI, developed by Anthropic, focuses on making models less harmful and more helpful by creating a “constitution” that outlines ethical principles and rules to guide the model [61]. Ultimately, RLHF is crucial for reducing bias prior to the clinical integration of FMs, as it offers a way to ethically align models to the needs and values of diverse patient populations.

### 3.2. Ensuring Trustworthiness and Reproducibility

Trustworthiness and reproducibility are paramount in the biomedical AI ecosystem, as they ensure the reliability and accuracy of AI models in critical translational and healthcare applications. As summarized in Table 2, this section provides an overview of the challenges in achieving consistent, reliable, and verifiable results in AI systems, as well as the evidence-based techniques we have identified to mitigate these issues. In Section 3.2.1, we discuss the essential data lifecycle concepts, highlighting how the Findable, Accessible, Interoperable, Reusable (FAIR) principles need to be supported by data integrity, provenance, and transparency in the current AI ecosystem. Section 3.2.2 delves into interpretability and explainability, emphasizing the need for AI/ML models to foster trust and understanding among users in translational settings. Section 3.2.3 covers enhancing AI accuracy, exploring model alignment strategies, and human-in-the-loop approaches to improve the performance and ethical alignment of FMs. Finally, Section 3.2.4 discusses algorithmic transparency, which is indispensable for both reproducing results and establishing trust in a model.

Table 2. (Abbreviations: FAIR—Findable, Accessible, Interoperable, Reproducible). We explored the challenges in the ability of AI systems to consistently produce reliable and verifiable results that instill confidence in their predictions and decisions in the biomedical field. Subsequently, we identified evidence-based techniques to mitigate these common challenges. A more detailed explanation of these concepts and relationships is described in Section 3.2.1, Section 3.2.2, Section 3.2.3 and Section 3.2.4.

#### 3.2.1. Essential Data Lifecycle Concepts

FAIR principles, which promote good practices for scientific data and its resources, have a long-standing foundation for establishing the trustworthiness and reproducibility of biomedical data, particularly regarding the discovery and reuse of digital objects throughout their lifecycle [62,63,64]. We underscore the critical role of data provenance in ensuring that biomedical AI complies with the principles of FAIR by offering a transparent and traceable record of data origins, processing, and usage. Since FMs are trained on datasets from various sources and increasingly across different modalities, tracking data provenance becomes essential for building trust and accountability among model developers, data creators, policymakers, and the public [65]. Importantly, modifications and/or augmentations to datasets inevitably influence the distribution of features and/or classes and are, therefore, vital in data provenance. These alterations may also introduce errors, noise, inconsistency, and discrepancy in smoothness and dynamics in comparison to real-world data, which ultimately impact the trustworthiness of model output [15,77]. Furthermore, preserving data lineage becomes arduous due to changing data sources, intricate workflows or data transformations, and alterations in schema. To address these challenges, Longpre et al. constructed a unified data provenance framework that builds on existing standards [1]. Importantly, the authors highlight the need for “codifying” data lineage through symbolic attribution and convey the importance of modality-agnostic frameworks that accommodate evolving metadata types and jurisdiction-specific requirements [65].

#### 3.2.2. Interpretability and Explainability

AI-augmented biomedical decisions are often met with concerns about the technology’s ability to explain how it arrives at specific outcomes. A model that can be understood in a human-friendly way is therefore essential to foster trust and engagement among clinicians and biomedical investigators. The goal of interpretability and explainability in AI is to meet this need, but widespread adoption is hindered by the challenge of finding techniques that are universally applicable, resource-efficient, and not prone to misinterpretation by humans. Interpretability in AI refers to the degree to which a model’s inner workings are comprehensible to users within the context of the application domain [78]. On the other hand, explainability is about describing the behavior of an AI model in human terms by highlighting potential influences of input features with the model output at a local or global level [78]. In a systematic review of explainability and interpretability in medical AI, the authors categorize the techniques into two approaches—a priori and a posteriori—helping to clarify the differences between these concepts [66]. A priori techniques are implemented during the model development phase with the aim of improving the model’s interpretability. Examples of a priori techniques include (a) selecting features that align with established and relevant biomedical concepts, (b) implementing regularization to penalize large weights, and (c) using models with simpler topologies [66,67,68]. A posteriori techniques are applied after the model has been trained, serving as methods to explain the model’s predictions with respect to the input. Examples of a posteriori techniques include (a) feature perturbation to monitor how slight changes in the input affect the model output and (b) counterfactuals that elucidate a model’s reasoning through the lens of “what if” scenarios [66,69,70]. This distinction offers important insight into how interpretability and explainability can be strategically approached at different stages of AI development, balancing design choices with post hoc explanation tools.

#### 3.2.3. Enhancing AI Reliability with Human-in-the-Loop

A human-in-the-loop approach to AI systems aims to solve complex tasks by integrating human expertise/insights into the decision-making loop [79] benefit from the rich knowledge and oversight of human experts [73]. Human-in-the-loop approaches can play a pivotal role in developing accurate and reliable biomedical AI applications such as medical diagnostics, treatment planning, and personalized medicine [72]. In these biomedical applications, human-in-the-loop approaches aim to leverage domain-specific knowledge to complement the data-driven nature of AI models and mitigate the risk of spurious correlations. In biomedical imaging, for instance, human experts can identify regions of interest from rich but sparse imaging data. While AI aims to automate this task across large datasets, the expectation is that it must perform at least as well as human experts and be able to distinguish between relevant signals and noise. This highlights the need to first examine how human experts are trained and how this process can be emulated in the training of AI models. When training a human expert to interpret medical imaging, a mentor typically presents a variety of cases covering different conditions. Through this process, trainees learn to distinguish between normal and abnormal findings, as well as between signal and noise, gradually developing the ability to make accurate diagnoses. Gupta et al. present a methodology for AI-driven microscopy analysis that demonstrates how expert guidance can effectively refine training data to closely mirror the human learning process [71]. Their approach enables experts to iteratively select and approve training data, allowing the AI model to first be trained on a large, noisy dataset and then fine-tuned on a curated, high-quality dataset with less noise and more meaningful signal. This expert-guided process enhances the model’s generalizability across diverse datasets and fosters confidence among users of the AI system. While incorporating human input requires caution to avoid introducing implicit biases, a structured approach that integrates diverse perspectives can avoid this harm.

#### 3.2.4. Transparency and Reproducibility

Transparency is a nuanced concept used in various scientific disciplines, but recently, it has been at the forefront of discussions pertaining to global AI regulation. In a 2019 study, “transparency” was found to be the most commonly used principle addressing AI ethical guidelines, and it is one of the five key ethical principles promoted in 84 AI studies [74]. It is imperative to note that transparency is recommended for both the technical features of an algorithm and practical implementation methodology within current social parameters [75]. Closely related to transparency, reproducibility encompasses methods such as sharing open data and open-source code to ensure that scientific findings are accurate, reliable, or otherwise reproducible [79]. As FMs rapidly expand in biomedical applications, standardizing AI transparency practices becomes essential for responsible clinical integration. Bommasani et al. propose a comprehensive framework of 100 transparency indicators for “Foundation Model Transparency Reports”, covering the entire supply chain of FMs, from upstream resources (data, labor, compute) to model properties (evaluations, capabilities, limitations) and downstream use and impact [77]. They advocate for the institutionalization of these transparency reports early in the industry’s development to reduce compliance costs, improve risk management, and establish stronger transparency norms [77]. While challenges such as the complexity of AI models, protection of proprietary information, rapidly evolving technologies, and the documentation of implicit biases and decision-making processes present hurdles, they also offer opportunities for innovation and improvement in AI documentation practices.

### 3.3. Safeguarding Patient Privacy and Security

In this section, we explore critical aspects of safeguarding patient privacy and security within the biomedical AI ecosystem. As summarized in Table 3, we identify recurring challenges related to the protection of patient data and outline evidence-based techniques to address these issues. In Section 3.3.1, we discuss strategies to maintain data security and ensure proper data provenance. In Section 3.3.2, we examine how technologies like blockchain, edge computing, and federated learning can enhance data security and privacy within cloud and hybrid cloud infrastructures. In Section 3.3.3, we address the risks and strategies to combat patient re-identification and membership inference attacks. In Section 3.3.4, we examine the challenges of memorization in AI models and discuss strategies to mitigate these issues. These sections collectively highlight the ongoing challenges of patient privacy within the realm of biomedical AI, emphasizing the importance of robust security measures and ethical practices.

Table 3. We explored recurring challenges regarding the maintenance of privacy and security of patient data. Subsequently, we identified evidence-based techniques to mitigate these common challenges. A more detailed explanation of these concepts and relationships is described in Section 3.3.1, Section 3.3.2, Section 3.3.3 and Section 3.3.4.

#### 3.3.1. Essential Data Lifecycle Concepts

In safeguarding access to sensitive patient data, it is essential to employ robust authentication methods, implement role-based access control, and maintain comprehensive logs of all data access instances [80]. However, while these mechanisms are fundamental for protecting patient data privacy, they are not without their challenges. For instance, role-based access control can become cumbersome in environments with complex permission hierarchies, and the sheer volume of data access logs can make it difficult to detect unauthorized activities effectively in large-scale systems. As these challenges illustrate, the complexity of managing data access is closely tied to the broader responsibility of ensuring that, as the need for data sharing and reuse in biomedical research grows, the rights and privacy of participants are respected and upheld. This process necessitates obtaining informed consent for data sharing, de-identifying data sets before dissemination, and ensuring that the data are only utilized for the purposes explicitly agreed upon [81,82]. Although informed consent is a cornerstone of patient privacy protection, it presents challenges of its own, such as ensuring that participants fully understand the complexities of how their data might be used and keeping consent documents relevant in the face of rapidly advancing technologies.

Moreover, fostering a secure and responsible AI ecosystem hinges on the effective implementation of data provenance. Data provenance involves the meticulous documentation of a dataset’s origin, movement, and transformations throughout its lifecycle. Such documentation not only enhances transparency but also aids in identifying and analyzing potentially malicious activities [83]. Nonetheless, the methods used to establish data provenance come with their own set of limitations, including the complexity of tracking data lineage in large-scale systems, the potential for inaccuracies in provenance data, and the difficulty of maintaining up-to-date records as data evolve. Collectively, these principles of data access control, informed consent, and data provenance play a critical role in ensuring the trust and safety of all stakeholders within the biomedical AI ecosystem.

#### 3.3.2. Protecting Patient Privacy in Cloud Storage and Computation

The advantages of developing AI in the cloud include access to a flexible hardware infrastructure specifically designed for AI. This infrastructure is equipped with state-of-the-art GPUs that not only accelerate the training process but also efficiently handle the influx of inference processing associated with the deployment of a new AI system. In this scenario, the trained neural network is put to work for practical applications. Furthermore, the cloud eliminates the need for complex hardware configuration and purchase decisions, providing ready-to-use AI software stacks and development frameworks. Cloud-based AI development also has its disadvantages. One of the primary concerns is the rising costs associated with storing large datasets and training models. Additionally, data security becomes a significant challenge. Ensuring the security and privacy of sensitive data during both storage and computation is crucial. This challenge is further amplified by the need to perform complex computations on this sensitive data without compromising its integrity. Therefore, while cloud-based AI development offers numerous benefits, it also presents complex challenges that need to be effectively addressed.

Data privacy is a critical issue when deploying models for clinical practice in cloud environments due to the sensitive nature of patient health information (PHI). This sensitivity restricts data storage infrastructure, network data transfers, and access to computing resources [89]. A hybrid cloud model enables organizations to utilize their own infrastructure for sensitive, private data and computation while integrating public clouds for nonsensitive, public data and computation [86]. Alternatively, a blockchain-based interplanetary file system (IPFS) can be implemented as secondary storage to safeguard the privacy and security of patient health information [84]. IPFS allows users to host and receive content in a decentralized manner, while blockchain ensures that once data are recorded, they cannot be altered without the consensus of the network. Blockchain technology securely transmits and stores patient data, providing a potential solution for addressing privacy concerns associated with rapid medical data access and processing. However, a ‘51% attack’ is a potential security risk in blockchain networks where a miner or group of miners, possessing over half the network’s computing power, can rewrite the blockchain [104].

Edge computing is a paradigm that shifts data processing and storage from the cloud closer to the source devices, enhancing data security and privacy while also decreasing latency [105]. It, therefore, offers utility in enhancing the security of patient-sensitive data within the biomedical AI ecosystem by preventing public data leaks while also speeding up decision-making, reducing latency, and improving the overall quality of care [106]. For example, Humayun et al. introduced a framework to integrate cutting-edge technologies like mobile edge computing and blockchain to enhance healthcare data security [107]. Meng et al. presented an edge computing-based approach for healthcare applications that store and perform computation on cloud servers [107]. They employed homomorphic encryption, which allows computations to be performed on encrypted data without the need for decryption, thus ensuring patient data security even if accessed by attackers [85]. They also presented a strategy to distribute computation across multiple virtual nodes at the edge, leveraging cloud computational resources while keeping all arithmetic operations masked. The approach prevents adversaries from discerning the specific tasks performed on the encrypted patient data. Serverless edge computing represents an evolution of cloud serverless technology, where there is an abstracting of servers from the application development process, enabling developers to build applications without concern for the underlying infrastructure [88]. Serverless edge computing highlights the potential for distributing models with preserved privacy, combining the flexibility of cloud computing with the security of local deployment [89]. However, while an edge or serverless edge computing approach offers a promising solution for enhancing patient data security, there is an added complexity of managing distributed systems.

Federated learning is a framework for distributed machine learning whereby patient data stored across various hospitals and healthcare institutions are kept decentralized on their local servers. These institutions use a federated workflow where learning takes place locally on their own nodes/edge devices. A central cloud server then aggregates the results to create a unified model. Sadilek et al. applied modern and general federated learning methods that explicitly incorporate differential privacy to clinical and epidemiological research [90]. They demonstrated that federated models could achieve similar accuracy, precision, and generalizability as standard centralized statistical models while achieving considerably stronger privacy protections. While federated learning enhances patient data privacy by decentralizing data processing, it faces challenges related to maintaining data integrity across multiple nodes.

#### 3.3.3. Patient Re-Identification and Membership Inference Attacks

Patient data are essential for developing FM models in biomedicine, but robust methods are needed to protect patient confidentiality. The first crucial step to address privacy concerns involves anonymization or removal of patient-identifiable information. The Department of Health and Human Services has specified the 18 types of protected health information to be removed from patient data in order to comply with the Health Insurance Portability and Accountability Act (HIPAA) [91]. In addition, rule-based and machine learning-based systems have been developed to de-identify/anonymize health data, including novel methods based on the self-attention mechanism [92,93,108]. However, de-identification strategies are proving to be insufficient in protecting patient records in the face of algorithms that have successfully reidentified such data [109,110]. Narayanan et al. showed that de-anonymization attacks could be highly effective even when the adversary’s background knowledge is imprecise, and the data have been perturbed prior to release, meaning an adversary with minimal knowledge about an individual can identify this individual’s record in a dataset [111]. This introduces us to a similar concept known as membership inference attacks (MIA), which is aimed at discovering whether a specific individual’s data were used in the training set of a model. Sarkar et al. demonstrated that de-identification of clinical notes for training language models was not sufficient to protect against MIAs [112]. Therefore, MIAs pose significant risks in exposing the personal information of individuals whose data contributed to a model, and strategies to mitigate this risk must be implemented. The general principle of MIA involves analyzing model responses to inputs that infer training data membership, revealing a model’s privacy vulnerabilities in addition to its potential overfitting or insufficient generalization [96,113,114,115]. Since MIA and protection against it has largely been studied on simple classification models, Ko et al. studied MIA strategies on multi-modal FMs trained on imaging and text data (CLIP). In their exploration of well-established MIA defense strategies for simple models applied to a multi-modal model, there were two important findings: (a) L2 regularization, a strategy to penalize the weights of the model to encourage smaller or sparser coefficients, moderately protects against privacy attacks by reducing model sensitivity to variations in input data; however, this comes at a cost to the model’s utility; (b) data augmentation, a technique that artificially applies various transformations to the data to increase their size and entropy (e.g., rotating or scaling an image), enhanced protection of the model against MIA (the AUC score for the weakly supervised attack went from 0.7754 to 0.7533) and improved the model’s generalizability (zero-shot performance improved by 1.2%) [94]. These findings support the idea that defense mechanisms that curb model overfitting also reduce the model’s susceptibility to MIA [95]. However, this privacy protection must be balanced against a potential utility penalty for the model [116]. Zhang et al. assessed the vulnerabilities of models trained on synthetic health data to MIA [117]. They studied MIA attacks on two types of synthetic data: (a) Full synthesis, which involves learning a generative model that mimics the real data distribution from which synthetic data are sampled while severing direct links to real individuals, and (b) partial synthesis, which employs a transformation function that perturbs features of real records to generate synthetic counterparts but maintaining some connection to the original data. The findings suggest that partial synthesis is more susceptible to membership inference attacks compared to full synthesis, indicating that the method chosen for synthetic data generation largely affects data privacy. Consequently, a full synthesis approach seems to be the optimal choice when training models on synthetic data. As previously discussed, synthetic data quality can be undermined by inaccuracies, noise, over-smoothing, and inconsistencies relative to real-world data, so it is crucial to ensure rigorous quality control measures when generating and using synthetic data [15].

Song and Mittal introduced a privacy risk score that was shown to align closely with the actual probability of a sample being from the training set; this is crucial for identifying which data points might be particularly vulnerable to MIA [97]. In a nutshell, the privacy risk score measures the likelihood that an input sample is part of the training dataset based on the observed behavior of the target machine learning model. This likelihood is defined by the posterior probability that a sample belongs to the training set given the observed outputs from the target model. However, the reliance on posterior probability can render the method sensitive to the model’s complexity and training dynamics, potentially diminishing its reliability in more intricate or less interpretable models. Despite these limitations, the privacy risk score remains a valuable tool, guiding decisions on model deployment in clinical settings and informing the development of stronger privacy preservation strategies before deployment.

#### 3.3.4. Memorization of Patient Data

The distribution of real-world data tends to be “long-tailed”, where a few categories contain most of the data, while a large number of categories have only a few samples [118,119]. Overparameterization in large AI models aids in capturing rare events at the “tails” of the dataset but introduces challenges like high computational costs, optimization difficulties, overfitting, performance loss when scaled down, and vulnerabilities to data leaks as well as manipulation [99,100]. For example, an FM trained on clinical data might memorize specific details about patients with a rare disease, increasing the risk that adversarial attacks could compel the model to reveal sensitive data despite precautions taken during model alignment [8]. Carlini et al. highlight the relativity of memorization by introducing the “Onion Effect”, where removing the most vulnerable outlier points reveals a new layer of data previously considered safe, now susceptible to privacy attacks [101,120]. Similarly, Hassan Dar et al. explored memorization in latent diffusion models used for creating synthetic medical images from CT, MRI, and X-ray datasets and found high levels of memorization across all datasets [98]. They observed that implementing data augmentation strategies can decrease the extent of memorization. Consequently, data augmentation appears to be an effective strategy for reducing memorization, as discussed in the previous section, and for enhancing protection against MIA. However, it is essential to fine-tune the data augmentation techniques to ensure that they do not compromise the performance of the model.

Deduplication of training data involves identifying and removing duplicate entries from a dataset. This task is particularly labor-intensive given the vast size of training datasets, which often span hundreds of gigabytes, rendering perfect deduplication impractical. Moreover, accurately matching patient records can present challenges due to inconsistencies in data entry and variations in patient information. Although generally reliable, there is a risk of losing important data during the deduplication process. Despite these limitations, deduplication remains a crucial strategy for managing patient datasets since duplicate records can lead to breaches of patient privacy as a result of data memorization. In fact, Carlini et al., in another paper on quantifying memorization in language models, demonstrated that sequences repeated fewer than 35 times see a statistically significant reduction in memorization from 3.6% to 1.2% with deduplication [101]. As a bonus, deduplication efforts also enable better model evaluation by diminishing train–test overlap and decreasing the number of training steps required to achieve the same or enhanced accuracy [101]. A few methods for deduplication of patient records include (a) exact substring duplication—when two examples share a sufficiently long substring, one is removed [102]; (b) suffix array—removal of duplicate substrings from the dataset if they occur verbatim in more than one example [121]; and (c) MinHash—an algorithm for estimating the n-gram similarity between all pairs of examples in a corpus and removing data with high n-gram overlap [122]. Deduplication of patient data requires a method for record linkage since directly comparing personal information across systems to identify duplicates violates privacy regulations and is not feasible with de-identified data [103]. Privacy-protecting linkage approaches of clinical data records first require the creation of secure and anonymous patient identifiers. Some approaches include the (a) U.S. NIH Global Unique Identifier (GUID)—which generates hash codes for personal identifiable information in records; (b) Mainzelliste—developed in Germany, it is an open-source service for pseudonymization that generates pseudonyms unlinked to identifiable elements but allows for data matching; and (c) European Patient Identity Management (EUPID)—generates context-specific pseudonyms using hashing algorithms and thus supports using different pseudonyms for the same patient in various contexts while assuring patient anonymity across the contexts [123]. These identifiers, such as the hash codes generated by GUID, can be compared to link and deduplicate patient data. In summary, deduplicating training data, although labor-intensive, can be essential for safeguarding patient privacy and maintaining the integrity of biomedical AI systems. By leveraging diverse techniques and privacy-preserving linkage methods, we can markedly decrease data memorization and ensure patient anonymity.

## 4. AI Stewardship

### 4.1. AI Governance and Regulation

The regulation of AI involves several key international and national bodies, each contributing uniquely to the governance landscape. To cultivate an environment where ethical AI practices such as the development of FMs flourish, governing bodies have concentrated on several key areas. First and foremost, transparency is emphasized by requiring clear and understandable AI decision-making processes. This ensures that the operations and outcomes of AI systems are accessible and comprehensible to all stakeholders. Additionally, accountability mechanisms are being established to hold AI developers and users responsible for their systems, thereby fostering trust and reliability. Concurrently, guidelines to mitigate biases and ensure fairness are being promoted and implemented, which work hand-in-hand with strict data-handling practices designed to protect individual privacy. A comprehensive governance approach supported and implemented by the federal and state are essential to ensure that AI development is both ethical and aligned with standardized societal values.

#### Landscape and Integration of AI Governance and Regulation

The European Union (EU) has been at the forefront of the AI governance initiative, implementing the General Data Protection Regulation (GDPR) in 2018, which sets stringent guidelines on data protection and privacy directly impacting AI development and deployment [124]. Building on this, the EU’s Artificial Intelligence Act, first introduced in 2021, aimed to create a harmonized framework for AI regulation, focusing on high-risk applications and promoting trustworthy AI practices. It is important to note that the European Commission’s High-Level Expert Group on AI (HLEG), established in 2018, also developed the Ethics Guidelines for Trustworthy AI, which emphasize the need for AI systems to be lawful, ethical, and robust [125]. Similarly, the Council of Europe’s Ad Hoc Committee on Artificial Intelligence (CAHAI), formed in 2019, is working towards a comprehensive legal framework for AI, focusing on protecting human rights, democracy, and the rule of law within the AI context [126]. Regarding international governance, the United Nations (UN) has also made significant strides through UNESCO’s Recommendation on the Ethics of Artificial Intelligence, published in 2021 [127]. This document serves as a global standard-setting instrument addressing human rights, ethical principles, and the need for transparency and accountability in AI contexts. Moreover, the World Health Organization (WHO) has also played a distinct role in the international governance of AI. In 2021, WHO released its first global report on AI in medicine, proposing six guiding principles of ethics and human rights: (a) Protecting Human Autonomy, (b) Promoting Human Well-Being and Safety and the Public Interest, (c) Ensuring Transparency, Explainability, and Comprehensibility, (d) Fostering Responsibility and Accountability, (e) Ensuring Inclusiveness and Equity, and (f) Promoting Sustainable AI [128]. Expanding on these six consensus principles, the WHO published a second report in 2023 on the practical implementations of AI systems in healthcare and biomedical science [129]. Importantly, the WHO is mindful of the growing need for capacity building and collaboration among different sectors/regions and is working to develop a global framework for the governance of AI systems for healthcare [124].

Meanwhile, in the United States (US), the Federal Trade Commission (FTC) and the National Institute of Standards and Technology (NIST) play crucial roles in AI regulation. The FTC’s Guidance on AI and Algorithms, issued in 2020, emphasizes the importance of fairness, transparency, accountability, and explainability for diverse stakeholders; it warns against biases and scientifically deceptive practices [130]. Similarly, the NIST’s Framework for Managing AI Risks, released in 2021, provides comprehensive guidelines to identify, assess, and manage AI-related threats, supporting the development of trustworthy and reliable AI systems [131]. Although not a direct result of the FTC’s and NIST’s work, both government agencies’ progress with documenting ethical AI practices contributed to the broader regulatory landscape culminating in the US government; it subsequently led the US Congress to enact the AI in Government Act of 2020. This act encourages federal agencies to adopt AI technologies while ensuring adherence to civil liberties, civil rights, as well as economic and national security [132].

Sharing the same vein as the aforementioned contributions to governance, the United States Executive Order on the Safe, Secure, and Trustworthy Development and Use of Artificial Intelligence outlines an up-to-date comprehensive framework to address these areas and push AI regulation via legal adherence [133]. The Executive Order (EO) is broken down into 16 clearly delineated sections and provides a clear, comprehensible, and accessible outline for diverse stakeholders. As previously mentioned, many of the regulatory contributions focus on a few domains (e.g., trustworthiness, privacy, and protection), but critically, the EO addresses a major gap in current frameworks as it encompasses a multifaceted call for ethical compliance within all domains of AI. Remarkably, it covers strategies to support workers in an AI-integrated economy, which has not typically been covered in other governance documentation in detail. Moreover, the EO has a detailed section defining mechanisms of implementation to support the adherence to the ethical guidelines discussed. Future regulatory bodies and contributions should follow the structure of the multifaceted framework and adapt the EO to implement their updated or more robust call for compliance. Overall, all these milestones reflect the ongoing efforts to standardize the safe and effective use of AI in biomedical applications, increase research funding to address ethical, legal, and social implications, as well as engage the public in discourse about AI’s role in healthcare.

Despite these collective efforts, several gaps remain in the current landscape of regulatory frameworks. A significant challenge includes the lack of global harmonization, leading to fragmented regulations that complicate compliance for international AI developers. Additionally, the rapid pace of AI advancements often outstrips the ability of regulations to keep up, necessitating more agile and adaptive regulatory mechanisms. Ethical guidelines also need to be more precisely defined and enforceable to effectively address issues such as bias and discrimination [134]. Finally, existing laws primarily focus on data protection and privacy, with insufficient attention to other ethical concerns like AI’s impact on employment, environment, and implementation. While significant strides have been made, ongoing efforts are needed to address existing gaps and keep pace with technological advancements. Altogether, global harmonization, the aggregation of current ethical considerations, the development of adaptive regulatory frameworks, and the lawful reinforcement of AI guidelines will be key to achieving these goals [135,136]. It is clear the regulation and governance of AI are crucial for ensuring an ethically grounded AI ecosystem, especially in sensitive fields like biomedical AI.

### 4.2. Stakeholder Engagement

Engaging with a diverse range of stakeholders is critical for optimizing ethical and responsible outcomes to successfully develop and deploy AI systems within the biomedical field (Figure 4). Stakeholder engagement involves identifying and interacting with all parties who are either affected by or can influence AI systems [137]. Stakeholders in the AI biomedical ecosystem can be broadly categorized into three levels: (1) individual stakeholders, (2) organizational stakeholders, and (3) national/international stakeholders [134].

Previous literature highlights a broad spectrum of stakeholders involved in the AI ecosystem, but there is minimal literature explaining the relevance of identifying stakeholders early in the AI lifecycle. Early identification allows AI system developers to discern which ethical guidelines are most pertinent to their products and services, aligning the development process with ethical standards from the outset [134]. It also helps in assessing who might be influenced by the AI systems and how to recognize individuals, groups, organizations, and even nation-states that could be affected or have the power to affect AI outcomes [134]. For example, investigators and developers are often more attuned to the technical and performance aspects of AI systems and are likely to express concerns regarding the ethical dimensions and impacts of AI decisions and activities. In comparison, non-expert stakeholders such as clinicians and general consumers contribute valuable insights into the real-world implications of AI systems, supporting responsible AI behavior in diverse contexts.

Early identification also helps us understand the specific concerns and needs of different stakeholders, in addition to bringing forth the concept of explainable AI (see Section 3.2.2)—a suite of machine learning techniques that enable human users to understand, appropriately trust, and produce more explainable models [138]. Users may lack the training to fully comprehend AI systems, which may lead to potential misuse or misinterpretation. This highlights the need for AI systems to provide clear and verifiable explanations of their decisions for relevant stakeholders. For instance, clinicians are frequently concerned about privacy breaches, personal liability, and the loss of oversight in clinical decision-making. Additionally, certain demographic groups may be disproportionately affected by AI systems based on factors such as region, age, socioeconomic status, and ethnicity [19]; therefore, special attention is needed to safeguard their interests. Moreover, the explanations provided to end-users might differ from those required by other stakeholders, emphasizing the importance of tailoring communication to the audience. To enhance widespread trust and accountability, AI systems must cater to the explanation needs of various stakeholders.

#### Co-Design

Implementing co-design principles in AI use and development encompasses actively involving stakeholders in the design process, ensuring their needs and concerns are addressed from the beginning of the AI pipeline. Co-design supports collaboration between AI developers, users, and other stakeholders, leading to more inclusive and ethically grounded AI systems [139]. By incorporating feedback from a diverse range of stakeholders, AI systems can better align with societal values and ethical standards, enhancing their acceptance and effectiveness. Co-design consists of an iterative process composed of designing, testing, and refining both hardware and software components until the system meets desired performance requirements [139]. This process bridges the gap between hardware and software design, in addition to AI deployment, which traditionally has been developed independently. Critically, co-design also supports human-in-the-loop (HITL) learning by fostering user engagement, ensuring systems are user-centric, and facilitating continuous feedback and improvement [73]. Key concepts in co-design include engaging end-users/diverse stakeholders throughout the AI development process, exposure analysis, and implementing ethical frameworks within design processes [140].

We have stated the benefits and significance of co-design and that this approach ensures diverse perspectives are incorporated into the AI system, leading to more robust and ethically aligned outcomes. It is imperative to note that early engagement directly relates to co-design. Engaging end-users and other stakeholders throughout the design process is crucial for capturing a wide array of needs and potential impacts, thus ensuring that the AI system is designed with a comprehensive understanding of its real-world application. This engagement helps in identifying potential ethical and practical issues early on, allowing for timely adjustments and improvements before AI applications are utilized by stakeholders [141]. Another critical component of co-design includes exposure analysis, which involves analyzing the extent to which different stakeholders are exposed to various aspects of the AI system and identifying potential risks [142]. By understanding the exposure levels of different stakeholders, developers can design safeguards and features that minimize risks and enhance the system’s safety and reliability. A critical component of exposure analysis includes evaluating potential contacts between hazards and receptors [143]. This involves assessing the interactions between potential hazards posed by the AI system and the stakeholders who might be affected by these hazards. By systematically evaluating these elements, developers can implement strategies to reduce stakeholder vulnerability, ensuring that the AI system is both safe and ethically sound.

A notable framework composed of multifaceted metrics for ethical practice in co-design includes Z-inspection, coined by Zicari et al., which focuses on evaluating and auditing AI systems at multiple stages of the AI pipeline [143]. Z-inspection involves a multidisciplinary approach, where ethicists, domain experts, and diverse stakeholders collaborate to inspect and assess the AI system at various stages of its development [143]. This inspection process helps in identifying potential ethical issues early on in AI development, allowing for timely interventions and modifications to be embedded throughout the AI pipeline. Crafting and implementing frameworks such as Z-inspection supports transparency and accountability across all stakeholders throughout AI development and cultivates an environment where ethical standards are continuously monitored and upheld, thereby enhancing the trustworthiness and reliability of an AI system. In conclusion, the co-design approach, coupled with ethical methodologies such as Z-inspection, plays a pivotal role in developing ethically grounded biomedical AI ecosystems. By active collaboration amongst stakeholders and continuous evaluation of ethical implications, we can create AI systems that are more aligned with societal values, ultimately leading to broader acceptance and enhanced outcomes in biomedical applications.

## 5. Concluding Unified Perspective

We have extensively reviewed the ethical challenges and the relevant mitigation strategies required to minimize the negative impacts of AI within clinical translation. These strategies act as checkpoints throughout AI development and deployment to maintain an ethical and trustworthy AI lifecycle. Additionally, we have examined regulations and recommendations globally for the responsible use of AI and emphasized the importance of engaging all relevant stakeholders during the development and integration processes of AI (see Table 4). With this comprehensive approach and with a unified perspective, AI technologies will meet the needs and protect the interests of all relevant stakeholders based on their specific interaction with the AI technology, fostering iterative improvement and adaptability of the entire ecosystem. Collectively, these elements form a robust Ethical and Trustworthy Artificial Intelligence (ETAI) Biomedical Ecosystem (Figure 5).

Table 4. We reviewed similar papers that addressed themes and practices related to implementing FMs in clinical practice, noting the key components covered by each paper. Our manuscript, listed at the end of the table, comprehensively addresses all these components, highlighting our holistic approach to building an ETAI Biomedical Ecosystem.

### 5.1. AI Lifecycle and AI Pipeline in the AI Biomedical Ecosystem

Understanding the distinction between the AI pipeline and the AI lifecycle is foundational for building an ethical and sustainable biomedical AI ecosystem. While both concepts are integral to the advancement and maintenance of AI systems, they conceptualize the design, development, deployment, and stewardship of the AI system distinctly. The AI pipeline represents a technical assembly line, where the creation and deployment of an AI model follows a linear, step-by-step process [145], starting from data collection and preprocessing, moving through from co-design, model training, and validation, and culminating in deployment; the pipeline’s primary goal is to deliver a functional AI system ready for clinical translation. Each component in the pipeline is clearly delineated, focusing on the efficiency and precision of the output [145]. Critically, the AI lifecycle embodies a dynamic, cyclical flow of AI system processes, emphasizing continuous iterative improvement and adaptation [144]. Rather than a straight path to an endpoint, the lifecycle views it as an ongoing journey, where the AI model system evolves through iterative feedback and refinement from all stakeholders. This approach integrates constant input from the developer and the user at every stage—from initial development through deployment and ongoing operation—promoting systems that remain relevant, ethical, reliable, and vibrant over time.

An important aspect of the AI lifecycle is the continuous evaluation of AI models to ensure safe, reliable performance in real-world biomedical applications. Standardized model testing, including the use of adversarial inputs, is essential to identify and mitigate potential vulnerabilities that could lead to unreasonable outputs or even system crashes. These adversarial inputs, intentionally designed to exploit weaknesses in the model, present a significant challenge in AI systems. Incorporating robust testing frameworks during model evaluation, including stress tests with adversarial data, can help ensure the resilience of AI models against such threats. Furthermore, model evaluation must also include continuous monitoring and fine-tuning post-deployment to safeguard the system from emergent risks.

In a nutshell, the AI pipeline ensures each model is constructed by addressing all functional steps efficiently and effectively, whereas the AI lifecycle guarantees that the model continues to comply with ethical standards and meet operational demands throughout its existence. Collectively, they form a comprehensive framework for our community and all stakeholders that support the development and maintenance of a robust and ethical biomedical AI ecosystem.

### 5.2. Standardizing Bias Mitigation, Trustworthiness and Reproducibility, and Privacy and Security

Mitigating bias, enhancing trustworthiness and reproducibility, and ensuring privacy and security are core components of ethical AI practices that must be integrated into each phase of the AI lifecycle to create a robust biomedical AI ecosystem [9,10]. As we have discussed, there are several methods to support the implementation of these ethical considerations. Briefly, enhancing trustworthiness and reproducibility requires transparent methodologies and documentation, enabling AI systems to be understandable and verifiable [77,79]. Trust in AI systems is built through both explainability and reproducibility. Additionally, privacy and security involve robust data protection practices, safeguarding patient information from unauthorized access and breaches throughout the AI lifecycle [80]. Privacy should be maintained in the initial data handling, as well as throughout the system’s operational life, adapting to new threats and vulnerabilities as they arise. To adapt to the rapid advancement of AI, these concerns must be addressed through standardized metrics and evaluation processes that are consistently applied across all stages of the AI lifecycle [80,81,82,83,84,85,86,87,88,89,90]. Standardization ensures that ethical considerations are not an afterthought but are integral to the design, implementation, and operation of AI systems. This systematic AI lifecycle approach allows for the continuous improvement and adaptation of AI technologies, ensuring they remain ethical, reliable, and align with societal values.

### 5.3. Call for Continuous AI Stewardship and Harmonious AI Governance in the AI Lifecycle

The development and deployment of AI systems in biomedicine must be accompanied by continuous AI stewardship and harmonious AI governance. Continuous engagement with diverse stakeholders, including patients, clinicians, ethicists, and policymakers, is foundational. This iterative engagement will ensure that AI systems are developed and deployed in ways that meet the needs and values of all affected parties [137]. Transparency throughout the lifecycle should ensure that all relevant stakeholders distinctly understand the FMs’ development process, the data used to train them, and the potential ethical and performance limitations that may arise [77]. Providing ongoing training for diverse stakeholders and raising awareness about ethical AI practices is imperative to establish iterative feedback loops for the continuous monitoring and refinement of AI model systems throughout the AI lifecycle. Multidisciplinary governance frameworks that are adaptable to emerging technologies and evolving societal norms are also needed to certify that AI systems are developed and used in ways that are safe, fair, and transparent. Developers and investigators should adhere to global regulations, such as the EO and the WHO AI Global Report, to ensure AI systems comply with legal standards for privacy, data protection, and ethical use. Harmonizing these regulations across jurisdictions can simplify compliance and promote global standards. Critically, governance organizations and policymakers should collaborate and use standardized practices to enhance policymaking processes informed by diverse perspectives and expertise. In conclusion, implementing AI stewardship and harmonious AI governance in the AI lifecycle is essential for sustaining an ethically grounded AI biomedical ecosystem. This unified perspective fosters trust and promotes innovation, ultimately improving clinical outcomes.

## Figures and Tables

**Figure 1 bioengineering-11-00984-f001:**
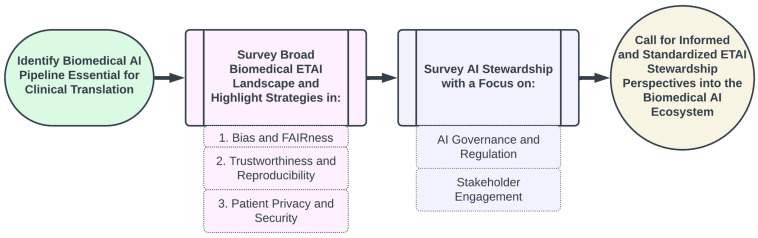
Contributions and outline. The paper first outlines and explains the biomedical AI pipeline, followed by an exploration of the Biomedical ETAI landscape. We then examine AI stewardship, culminating in a call for the adoption of standardized ethical practices and unified perspectives.

**Figure 2 bioengineering-11-00984-f002:**
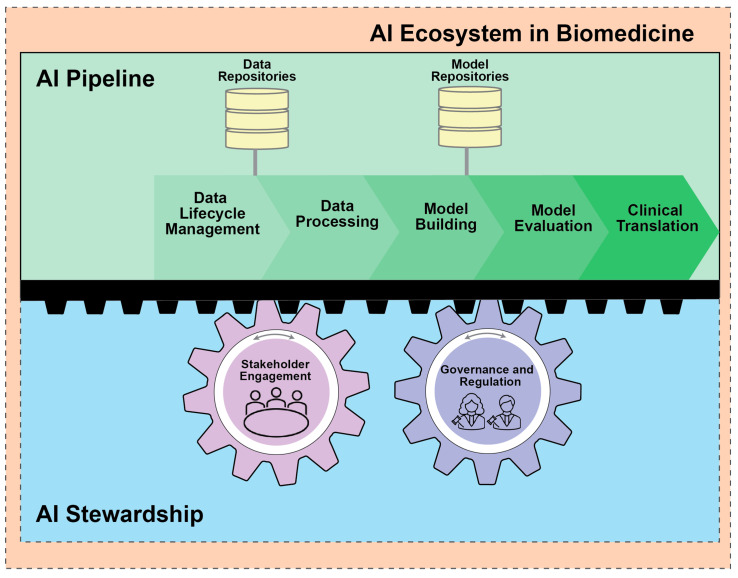
AI ecosystem in biomedicine. The AI pipeline begins with “Data Lifecycle Management” and concludes with “Clinical Translation”, with data and model repositories playing a critical role in sustaining the pipeline progression in their respective areas. Notably, AI stewardship, such as AI stakeholder engagement and governance regulation, influence the pipeline’s progression bidirectionally.

**Figure 3 bioengineering-11-00984-f003:**
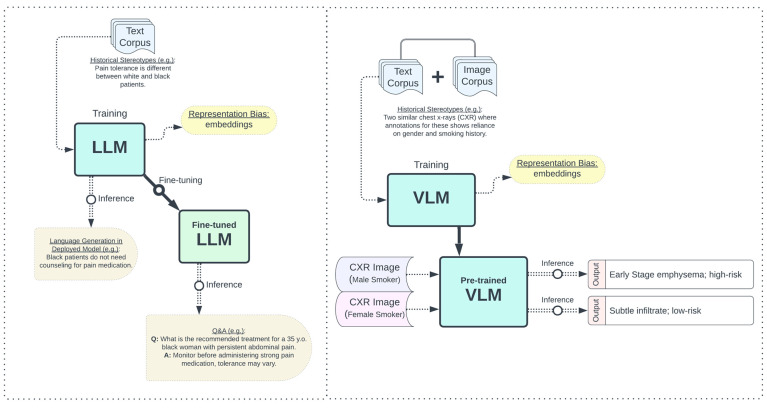
Impact of historical stereotypes on bias propagation in LLMs and VLMs for biomedical applications. This figure illustrates how historical stereotypes embedded in natural language and image datasets can propagate biases throughout model training and inference. Through biased embeddings formed during training, models may generate harmful inferences in clinical settings, such as recommending inadequate pain management for Black patients or inaccurately assessing risk levels in medical imaging based on gender or smoking history. The figure emphasizes the potential for biased data to shape clinical outcomes, underscoring the ethical importance of addressing representation bias in biomedical AI models.

**Figure 4 bioengineering-11-00984-f004:**
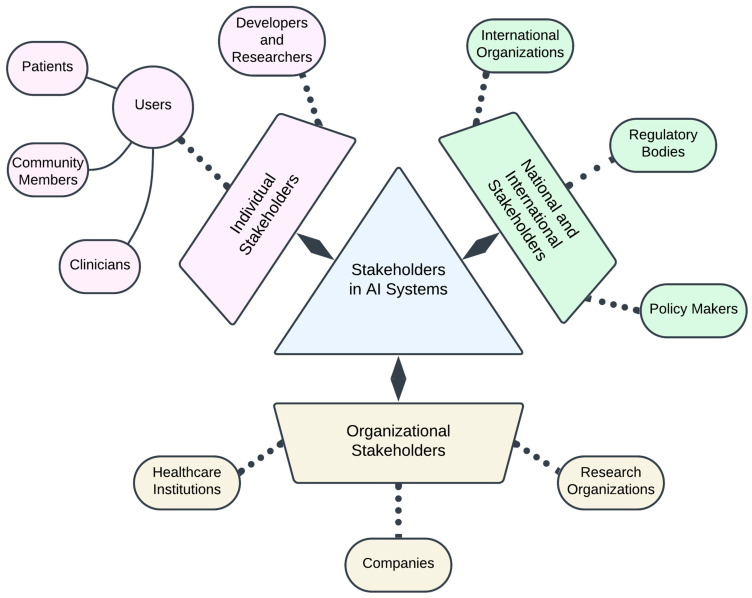
Stakeholder engagement. Individual stakeholders include users, developers, researchers, and any other individuals directly interacting with or impacted by AI systems. Users encompass clinicians and other non-expert individuals who bring a real-world perspective on the responsible use of AI. Organizational stakeholders are entities such as healthcare institutions, research organizations, and companies involved in the development, deployment, and maintenance of AI systems. National and international stakeholders encompass regulatory bodies and policymakers engaged in crafting laws and regulations governing AI technologies.

**Figure 5 bioengineering-11-00984-f005:**
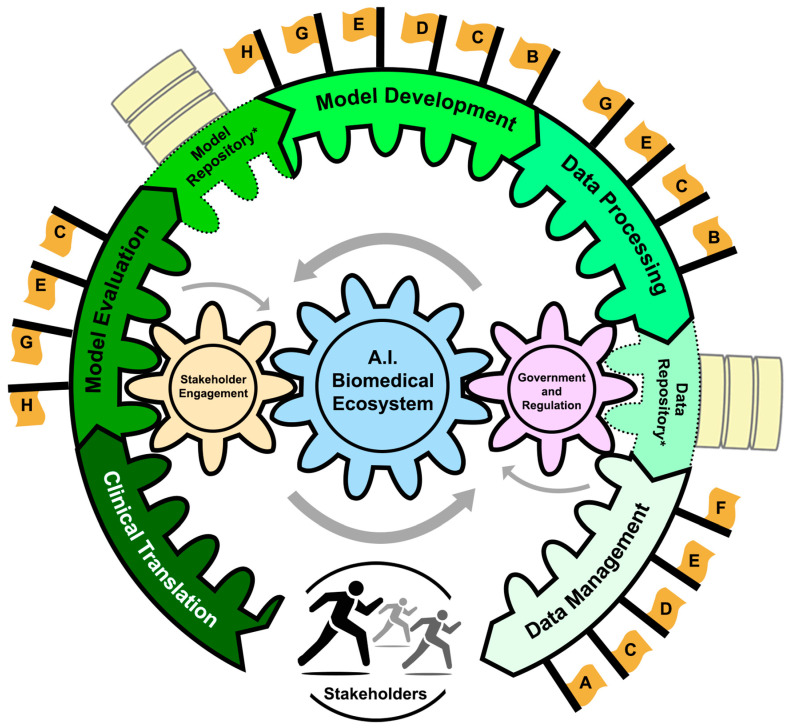
AI lifecycle. The figure illustrates the traditional AI pipeline as a cyclical system. Stakeholder engagement and government regulation act as gears that drive and shape this cycle. The bidirectional arrows signify that stakeholder and regulatory input can either propel the cycle forward or lead to revisiting earlier stages for refinement. This dynamic interaction enables continuous feedback between data management, model development, evaluation, and clinical translation, ensuring that these processes inform each other iteratively. As a result, the system fosters the development of AI technologies that are trustworthy, transparent, and fully aligned with ethical standards. Key checkpoints in the pipeline, marked with letters, denote significant milestones: A (securing data); B (combating data memorization); C (mitigating adversarial attacks); D (securing cloud infrastructure); E (mitigating demographic underrepresentation); F (data transparency); G (mitigating stereotype biases); and H (model transparency, interpretability, and explainability). These checkpoints ensure the AI system’s integrity, security, and efficacy, continuously aligning with evolving societal needs, ethical standards, and legal requirements.

**Table 1 bioengineering-11-00984-t001:** Mitigating social bias and enhancing fairness in biomedical AI.

Category	Strategy	Limitation/Challenge
Challenge: Underrepresentation of certain demographic groups (e.g., African Americans or women) in biomedical data can result in biased and unfair model decisions.		
Data Level (all dataset types)	Inclusive biomedical data collection [13]: - Ensuring all demographic groups are adequately represented. 2. Synthetic datasets [14,15]: - Data engineering to closely mirror phenotypes of underrepresented individuals.	Difficulty accessing diverse populations. Overcoming skepticism in certain demographic groups. Inaccuracies, noise, over-smoothing, and inconsistencies when compared to real-world data.
Training Level (datasets with labeled demographic metadata)	Importance weighting [16,17]: - Samples from underrepresented groups in the dataset are shown more frequently to the model, giving them higher importance. 2. Adversarial Learning [18,19]: - Train the primary model for the main task and an adversary to be unable to predict sensitive demographic attributes from the primary model’s output. 3. Regularization [20,21]: - Inclusion of fairness metric in the objective function to penalize unfairness. 4. Dropout [22]: - Probabilistic masking of neurons to reduce dependency on sensitive demographic features.	Overfitting to underrepresented instances. Optimization challenges due to simultaneous training of the primary model and adversary. Computationally expensive with large predictors. Requires careful tuning of the regularization parameter. Optimization challenges due to stochastic nature. Requires careful tuning of dropout rate.
Evaluation Level (datasets with labeled demographic metadata)	Equalized odds [23,24]: - Are TP and FP rates equal between demographic groups? 2. Equal Opportunity [25]: - Are TP rates equal for all demographic groups? 3. Predictive Parity [24,26]: - Is precision equal across demographic groups? 4. Explainability Methods [27,28]: - Assessing the model’s dependence on sensitive demographic attributes in decision-making.	1–3. Relying on a single fairness metric might give an incomplete or misleading picture of a model’s bias. 4. Application of techniques can be resource intensive. Effectiveness of explainability methods often depends on human interpretation.
Challenge: Human stereotypical biases can contaminate natural language data and affect the fairness of large language and vision-language biomedical models.		
LLM	Counterfactual data augmentation [29,30]: - Augment a corpus by swapping demographic identifier terms (e.g., swap he and she). 2. Bias control training [31]: - Learning to associate special control tokens with stereotypically categorized text to adjust model responses during inference accordingly. 3. Debiasing word embeddings [32,33]: - Generating augmented sentences through demographic identifier word swapping, encoding both original and augmented sentences and maximizing their mutual information. 4. Attention head pruning [34]: - Ablate subset of attention heads encoding stereotypes.	May fail to address deeper, more intricate biases that are not directly linked to demographic identifier terms. Bias control is a function of the categorization of responses with their stereotype associations. Can struggle with maintaining semantic consistency. Can potentially lose valuable context and meaning. Specific to attention-based architectures.
VLM	Additive Residual Learner [35]: - Disentangle skewed similarity in the representation of certain images and their demographic annotations.	The model may overcompensate and flip similarity skew with different demographic annotations.
Versatile (LLM and VLM)	Model Alignment [36,37,38,39]: - Techniques to tune AI systems to align with human preferences and values (e.g., reinforcement learning with human feedback).	Curating high-quality, ethically aligned output can be labor-intensive and time-consuming. Alignment can come with performance trade-offs.

**Table 2 bioengineering-11-00984-t002:** Ensuring trustworthiness and reproducibility in biomedical AI.

Category	Strategy	Limitation/Challenge
Challenge: The absence of standardized data management protocols throughout its lifecycle obstructs collaboration in biomedical AI, particularly in integrating diverse datasets for comprehensive model development.		
Biomedical Data	FAIR data principles [62,63,64] - Good practices for discovery and reuse of digital objects. 2. Data Provenance [65] - Documenting data origins, processing/transformations, and usage.	Broad implementation of FAIR principles is challenged by data fragmentation, interoperability issues, inadequate documentation, and the need for appropriate infrastructure and resources to implement effective data management practices. Maintaining data lineage can be cumbersome due to evolving data sources, complex workflows or data transformations, and schema modifications.
Challenge: The decision-making process and output of a biomedical AI model must be transparent, understandable, and verifiable by human experts to build the trust required for integration into clinical and translational settings.		
Biomedical AI systems	Interpretability and Explainability Methods [66,67,68,69,70] - Techniques incorporated into model development or evaluation to enhance understanding of the model’s inner workings and the impact of data attributes on decision-making. 2. Integrating Human Expertise [71,72,73] - AI leveraging human expertise and insights to improve its performance, learn from mistakes, and make more informed and accurate decisions. 3. Transparent AI Documentation [74,75,76,77] - Documentation of upstream resources used in development, model-level properties, and downstream applications.	Application of techniques can be resource intensive. Effectiveness of explainability methods often depends on human interpretation. Human input can introduce implicit biases. Complexity of AI models, proprietary information protection, rapidly evolving AI technologies, and the difficulty in documenting implicit biases and decision-making processes within the models.

**Table 3 bioengineering-11-00984-t003:** Safeguarding patient privacy and security in biomedical AI.

Category	Strategy	Limitation/Challenge
Challenge: Unauthorized access to sensitive biomedical data jeopardizes individual privacy, undermines stakeholder trust, and threatens compliance with regulatory guidelines.		
Data Life Cycle	Role-based access control, data access logs, and strong authentication methods [80] - Common data security measures to protect sensitive information. 2. Obtaining and respecting patient consent [81,82] - Informed consent; adhering to agreed terms; allowing withdrawal 3. Data Provenance [83] - Detailed audit trail of origin and journey of data, including their creation, movement, and transformation.	Managing complex permissions and challenges in detecting unauthorized activities in large-scale logs. Ensuring comprehension of complex data use scenarios and maintaining up-to-date consents amid rapidly evolving technologies. Complexity of tracking data lineage in large-scale systems, potential inaccuracies in provenance data, and the challenge of maintaining up-to-date provenance information.
Challenge: Developing biomedical AI in the cloud offers flexible, ready-to-use, and scalable infrastructure. However, ensuring the security and privacy of sensitive data during storage and computation is challenging in this environment.		
Data Storage	Blockchain technology [84] - Provides a decentralized, tamper-resistant ledger that ensures transparent and secure data recording. 2. Data Encryption [85] - Algorithms to transform readable data to unreadable data.	A ‘51% attack’: a potential security risk in blockchain networks where a miner or group of miners, possessing over half the network’s computing power, can rewrite the blockchain. Computationally intensive and may introduce latency issues.
Computation	Hybrid Cloud [86] - Flexible public and private cloud approach for optimizing security. 2. Edge Computing [87,88,89] - Allows for computing closer to data sources, reducing potential exposure during transmission. 3. Federated Learning [90] - Enables decentralized models to be trained across multiple devices or servers holding local data samples without exchanging them.	Integrating private cloud with public cloud introduces additional infrastructure complexity. Increased complexity of managing distributed systems. Maintaining data integrity across multiple nodes.
Challenge: Sensitive biomedical data are vulnerable to adversarial attacks, including the risk of re-identifying individuals or inferring whether an individual’s data were used to train a model.		
Patient Re-identification	Removing patient identifiers and/or implementing rule and ML-based patient data anonymization [91,92,93] - Manual or computational elimination of personally identifiable information from health data.	Algorithms have demonstrated the ability to successfully identify an individual’s record in a dataset.
Membership Inference Attacks	Techniques to increase model generalization [94,95] - Strategies to encourage smaller or sparser coefficients. 2. Data augmentation [95] - Techniques that artificially transform data to increase its size and entropy. 3. When training on synthetic biomedical data, employ a full synthesis approach [96] - Training a generative model to create synthetic data that mirrors the real data distribution. 4. Privacy risk score [97] - Measures the likelihood that an input sample is part of the training dataset.	Must be balanced against a potential utility penalty for the model. Must be balanced against a potential utility penalty for the model. Inaccuracies, noise, over-smoothing, and inconsistencies when compared to real-world data. Reliance on posterior probability can render the method sensitive to the model’s complexity and training dynamics.
Challenge: Large-scale biomedical models can memorize data, making them vulnerable to unique security risks such as data leaks, manipulation to produce misleading outputs, and the exposure of sensitive patient information.		
Over-Parameterized Models and “Long-Tailed” Data	Data augmentation [98] - Techniques that artificially transform data to increase its size and entropy. 2. Model size reduction [99,100] - Techniques to compress the model. 3. Patient data deduplication [101,102,103] - Identifying and removing duplicate entries from a dataset.	Must be balanced against a potential utility penalty for the model. Must be balanced against a potential utility penalty for the model. Appropriately matching patient records; potential loss of important data.

**Table 4 bioengineering-11-00984-t004:** Review comparison of key components in biomedical AI.

	Bias Mitigation Strategies	Data Privacy & Security	Fairness	Stakeholders and Governance	AI Lifecycle Ecosystem	Clinical Integration	Standardization of Ethical FMs	Biomedical Contexts/Applications
Fairness and Bias in AI: A Brief Survey of Sources, Impact, and Mitigation Strategies [9]	X		X	X				
The Evolutionary Dynamics of the AI Ecosystem [10]				X	X			
Bias and Fairness in LLMs: A Survey [25]	X		X					
Assessing the Research Landscape and Clinical Utility of LLMs: A Scoping Review [48]	X	X				X		
An AI Life Cycle: From Conception to Production [144]	X		X	X				
Building an Ethical and Trustworthy Biomedical AI Ecosystem for the Translational and Clinical Integration of Foundational Models	X	X	X	X	X	X	X	X

## Data Availability

No new data were created or analyzed in this study. Data sharing is not applicable to this article.

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
