# Peer review of "Building an Ethical and Trustworthy Biomedical AI Ecosystem for the Translational and Clinical Integration of Foundation Models"

_bioengineering, 2024, doi:10.3390/bioengineering11100984_

Round 1

Reviewer 1 Report

Comments and Suggestions for Authors

The authors present an Ethical and Trustworthy Biomedical AI Ecosystem for the Translational and Clinical Integration of Foundational Models. Some results are interesting and well-presented, which could make valuable contributions to the research community. However, I recommend some revisions on the following aspects of the manuscript.

-        Authors need to avoid unnecessary general information. They should focus on including important opinions and contributions of the study instead of general information.

-        Page 01, line 35: Repeated the abbreviation for FMs.

-        Page 01, line 38: Authors need to abbreviate the term in the text.

-        Need to check lines 52-54 on page 02.

-        Need to provide the flow chart for these lines 74-94 in the introduction section and highlight the novelty of the present work.

-        Rewrite the caption for Figure 1.

-        Need to cite the Table 1 in the proper place.

-        In Table 1, why did the authors repeat the abbreviations for LLMs and VLMs?

-        Page 09, line 309: Repeated the abbreviation for VLMs.

-        Page 09, line 318: Typographical errors are observed in this sentence. Please review it.

-        Authors need to add pictorial representations of the results related to stereotypical biases (Section 3.1.2).

-        Page 12, lines 419 and 426: How many times do authors abbreviate FAIR? On page 11, line 408: Authors have already abbreviated FAIR.

-        Need to add the caption for Table 3.

-        Why are there no references for sections 5, 5.1, 5.2, and 5.3? Please check it carefully.

-        This section should present the conclusion and outlook of this review.

-        Corresponding author email addresses are missing.

-        It is suggested to split the citations presented in a collective form to better justify the importance of each reference selected to present this topic.

-        To enhance the manuscript, the authors could compare their research with other studies in a table to highlight the benefits of their work.

-        Finally, the text should be checked for grammatical, formatting, and punctuation mistakes.

Reviewer 2 Report

Comments and Suggestions for Authors

It is a well-written comprehensive review paper that provides an overview of AI biomedical ecosystem with ethical and clinical data concerns.

The following comments could be considered to improve the manuscript:

1) In Section 3.1, the data bias was addressed. In fact, the data acquisition cannot involve a variety of biomedical data from demographic groups due to the limited sources and increasing cost. I think a possible solution to the challenge: the foundational AI model provider could address the data volume and demographic ratio information when issuing their model, and the users can use their own data to fine-tune these pre-training AI models. The authors could discuss about this.

2) In Figure 3, Model Evaluation could include standardized model testing to avoid the unreasonable output from AI models or prevent the systematic crash due to some adversarial data input. The context could discuss about this challenge and possible solutions in the AI biomedical ecosystem.

Round 2

Reviewer 2 Report

Comments and Suggestions for Authors

The authors have revised the manuscript by considering my review comments. The paper now is satisfied and could be considered for publication.